# Enhancing the emergency department experience for older adults: Study protocol for the implementation of a comfort menu and cart

Silvia A. Ferreira[1], Fernanda Bellolio[2,3], Susan M. Bower [2,4], Luis Fernando Penna [1], Magali A. L. Marion[1], Marlon R. Aliberti[1,5,6], Thiago J. Avelino-Silva[1,5,6,7], Christian V. Morinaga [1], Pedro Kallas Curiati [1,5]*

1 Geriatric Emergency Department Research Group (ProAGE), Hospital Sírio-Libanês, São Paulo, São Paulo, Brazil, 2 Department of Emergency Medicine, Mayo Clinic, Rochester, Minnesota, United States of America, 3 Department Health Science Research, Division of Health Care Policy and Research, Mayo Clinic, Rochester, Minnesota, United States of America, 4 Department of Nursing, Mayo Clinic, Rochester, Minnesota, United States of America, 5 Geriatric Center for Advanced Medicine, Hospital Sírio-Libanês, São Paulo, São Paulo, Brazil, 6 Laboratório de Investigação Médica em Envelhecimento (LIM-66), Serviço de Geriatria, Hospital das Clinicas, Faculdade de Medicina, Universidade de São Paulo, São Paulo, Brazil, 7 Division of Geriatrics, University of California, San Francisco, California, United States of America

* pedro.kcuriati@hsl.org.br

## Abstract

### Introduction

The aging of the population is a global phenomenon, with projections indicating a significant increase in the proportion of individuals aged 65 years and older by 2050. This demographic shift requires adapting emergency department (ED) services to meet the specific demands of older patients, who often present with multiple comorbidities and face challenges such as sensory and cognitive difficulties. EDs, traditionally designed for acute illness and injury management, may not be adequately equipped to meet the unique needs of this vulnerable population. This can result in suboptimal patient experiences, prolonged ED stays, increased hospitalizations, and poorer outcomes.

### Methods

This study protocol outlines a before-and-after study to evaluate the impact of implementing a comfort menu and cart on the experience and outcomes of older patients treated in the ED. The study will be conducted in the ED of Hospital Sírio-Libanês (HSL), a tertiary private hospital in São Paulo, Brazil. Patients aged 65 and older who presented to the ED will be eligible for inclusion. Participants will be recruited in two phases: pre-intervention and post-implementation of the comfort menu and cart. Data will be collected through patient and staff interviews, chart reviews, and a 30-day follow-up interview. Patient experience, staff experience, length of hospital stays,

**Data availability statement:** No datasets were generated or analysed during the current study. Following study completion, de-identified participant data and the full study protocol will be made available from the corresponding author upon reasonable request.

**Funding:** This research is currently funded by a Research Productivity Grant from National Council for Scientific and Technological Development (Conselho Nacional de Desenvolvimento Científico e Tecnológi, CNPq) awarded to Pedro K. Curiati (process no 305462/2024-5; approved July 24, 2025). Contact information: +55 61 3211-4000; atendimento@cnpq.br. The funder has no role in design, conduct, analysis, nor reporting of trial.

**Competing interests:** The authors have declared that no competing interests exist.

hospital costs, ED readmissions, falls, delirium incidence, quality of life, functional status, cognitive performance, and mortality will be assessed.

## Expected results

We expect to demonstrate the positive impact of implementing the comfort menu and cart in the ED on patient-centered outcomes. We anticipate improvements in the experience of older patients and medical and multidisciplinary staff, and hope to identify improvements in other exploratory outcomes.

## Trial registration number

ClinicalTrials.gov NCT06681376.

---

## Introduction

The aging of the population is a global phenomenon, with projections indicating a significant increase in the proportion of individuals aged 65 years and older by 2050 [1]. This demographic shift requires adapting emergency department (ED) services to meet the specific demands of older patients, who often present with multiple comorbidities and face challenges such as sensory and cognitive difficulties [2,3]. EDs, traditionally designed for acute care, may not adequately address the complex needs of this population, leading to suboptimal patient experiences and potentially poorer outcomes.

International organizations have recognized the need for specialized geriatric ED care and have published guidelines to improve the quality and safety of care for older adults in this setting [4,5]. The American College of Emergency Physicians, in collaboration with the American Geriatrics Society, the Emergency Nurses Association, and the Society for Academic Emergency Medicine, published the "Geriatric Emergency Department Guidelines" in 2014 [4]. They provide comprehensive recommendations on modifying the physical environment, training staff, and implementing specific care processes for older patients [4]. Similarly, the European Task Force on Geriatric Emergency Medicine published recommendations in 2020 emphasizing the importance of comprehensive assessment, including screening for delirium, depression, and fall risk [5].

Despite these guidelines, the implementation of evidence-based geriatric ED care practices remains inconsistent, and research exploring innovative interventions to improve the older patient experience is still in its early stages. A recent study by Lichen et al. (2021) demonstrated the positive impact of non-pharmacological interventions, specifically a "comfort cart," on the experience of older patients in the ED [3]. This study highlighted the potential of relatively simple, low-cost, patient-centered interventions, such as extra blankets or pillows, a personal hygiene kit, items for distraction, and items to enhance communication like reading glasses and hearing amplifiers, to enhance the comfort, well-being, and ability to communicate of older adults during their ED stay. Similar initiatives, such as the comfort care cart described by Stolzman et al. (2020) in the intensive care unit, have also shown promising

results in improving patient and family experience [6]. These studies suggest a growing recognition of the importance of addressing the holistic needs of older patients in acute care settings, beyond solely focusing on medical treatment.

Our study aims to build upon this emerging body of literature by implementing and evaluating the impact of a comfort menu and cart in the ED of a tertiary philanthropic hospital in Brazil. This will be the first study to explore this intervention in older adults in the ED setting since its original description by Lichen et al. [3], and it will be the first to generate evidence on its impact on patient-centered outcomes measured up to 30 days after the initial ED visit. By comprehensively assessing the impact of this intervention on patient experience, staff experience, and clinical outcomes, our study will generate valuable evidence to inform the development and implementation of this innovative non-pharmacological strategy to improve the care of older adults in EDs.

## Objectives

### Primary objective.

- Evaluate the impact of implementing a comfort menu and cart on the experience of older patients treated in the ED.

### Secondary objectives.

- Evaluate the impact of implementing a comfort menu and cart on the experience of medical and multidisciplinary staff involved in caring for older patients treated in the ED.

- Evaluate the impact of implementing a comfort menu and cart on other exploratory outcomes, such as length of hospital stay, hospital costs, and patient-centered outcomes up to 30 days in older adults treated in the ED, including readmissions, falls, delirium incidence, quality of life, functionality, and cognitive performance.

## Methods

### Study design

This study will utilize a quasi-experimental, before-and-after design [7]. Schedule of enrolment, assessments and intervention is summarized in Fig 1.

### Study population

#### Inclusion criteria.

- ED attendance at Hospital Sírio-Libanês (HSL), São Paulo, Brazil.

- Age ≥ 65 years.

- Capacity to consent and respond to the interview or the presence of a companion capable of doing so.

#### Exclusion criteria.

- Decline to participate in the study or use the comfort menu and cart.

- Absence of a companion able to consent to study participation and provide necessary information for patients with altered mental status or cognitive impairment.

- Decreased level of consciousness.

- Hemodynamic instability.

- Acute respiratory failure.

- Inability to be contacted by phone for the follow-up interview.

| Timepoint | Study Period | | | |
|---|---|---|---|---|
| | Enrolment | Pre-intervention Period | Post-intervention Period | Follow-up |
| | (t0-1) | (t0) | (t1) | (t2) |
| | *April-September 2026* | *April-June 2026* | *July-September 2026* | *Day 30 (± 2)* |
| **ENROLMENT** | | | | |
| Eligibility Screening (Inclusion/Exclusion Criteria) | X | | | |
| Informed Consent | X | | | |
| Allocation (by time period) | X | | | |
| **INTERVENTION** | | | | |
| Comfort Menu and Cart | | | X | |
| **ASSESSMENTS** | | | | |
| **Patient Interview (at ED visit)** | | | | |
| Patient Experience Questionnaire (adapted from McCusker et al. & Lichen et al.)[1] | | X | X | |
| Demographics | | X | X | |
| Source of Information | | X | X | |
| Sensory Deficits | | X | X | |
| History of Falls (past year) | | X | X | |
| **Clinical Assessments (at ED visit)** | | | | |
| Comorbidity (Charlson Comorbidity Index - CCI) | | X | X | |
| Risk of Falls (MEDFRAT & Carpenter instrument) | | X | X | |
| Frailty (Clinical Frailty Scale - CFS v2.0) | | X | X | |
| Clinical Severity (National Early Warning Score - NEWS 2) | | X | X | |
| Mood (Geriatric Depression Scale - GDS-15) | | X | X | |
| Geriatric Vulnerability (PRO-AGE Score) | | X | X | |
| Delirium Screen (DTS & bCAM) | | X | X | |
| Quality of Life (EQ-5D) | | X | X | |
| Functionality (Katz Index, Barthel Index, Lawton Scale) | | X | X | |
| Cognitive Performance (10-Point Cognitive Screener - 10-CS) | | X | X | |
| **Staff Interview (concurrent with patient recruitment)** | | | | |
| Staff Experience Questionnaire (adapted from Lichen et al.)[2] | | X | X | |
| **Chart Review & Administrative Data** | | | | |
| Length of Hospital Stay | | X | X | |
| Hospital Costs | | X | X | |
| **Follow-up Phone Interview** | | | | |
| ED Readmissions (since initial visit) | | | | X |
| Falls (since initial visit) | | | | X |
| Delirium Incidence (Family Confusion Assessment Method - FAM-CAM) | | | | X |
| Mood (GDS-15) | | | | X |
| Quality of Life (EQ-5D) | | | | X |
| Functionality (Katz Index, Barthel Index, Lawton Scale) | | | | X |
| Cognitive Performance (10-CS) | | | | X |
| Mortality | | | | X |

**Abbreviations:** bCAM, Brief Confusion Assessment Method; DTS, Delirium Triage Screen; ED, Emergency Department; EQ-5D, EuroQol 5-Dimension; FAM-CAM, Family Confusion Assessment Method; GDS-15, Geriatric Depression Scale (15-item); MEDFRAT, Memorial Emergency Department Fall Risk Assessment Tool; NEWS 2, National Early Warning Score 2; PRO-AGE, Predicting Hospital Admission and Prolonged Length of Stay in Older Adults in the Emergency Department.

[1] *Patient interview includes questions about comfort, specific care needs, respect, communication, and (in the post-intervention period only) use and perception of the comfort cart.*

[2] *Staff interview includes questions about patient comfort, provision of age-friendly care, and (in the post-intervention period only) perception and utility of the comfort cart.*

**Fig 1. Schedule of enrolment, interventions, and assessments (SPIRIT).**

**Sample size calculation.** The sample size calculation was based on the study by Lichen et al. (2021), which reported a difference of 0.98 on a Likert scale (ranging from 1 to 6) when assessing patient experience before and after the comfort cart intervention [3]. Considering a power of 90% and a significance level of 5%, 132 patients will be required in each group (pre- and post-intervention), totaling 264 participants.

## Study setting

The study will be conducted in the ED of HSL, a tertiary philanthropic hospital in São Paulo, Brazil [8]. The ED receives over 80,000 patients annually, providing continuous care for private and insured patients seeking emergency treatment. Patients undergo initial triage using a modified Manchester Triage System to classify risk and prioritize care [9]. In 2017, HSL inaugurated the Specialized Geriatric Emergency Department (ProAGE), which offers additional geriatric risk assessment for patients over 70 [10]. In 2019, the ED received the Geriatric Emergency Department Accreditation (GEDA), marking the first hospital in the Southern Hemisphere to achieve this recognition [11].

The ED has 14 examination rooms, 3 procedure rooms, 15 chairs for rapid medication administration, and 22 private rooms where patients requiring complex care due to age, mobility limitations, frailty, behavioral changes, or continuous vital sign monitoring are accommodated. Between January 2023 and June 2024, an estimated 9,608 patients over 65 years old were treated in these private rooms, averaging 533 patients per month.

## Data collection

**Recruitment.** A trained research assistant will recruit participants in the ED, working 25 hours per week (five 5-hour shifts between 10:00 am and 10:00 pm) over six months (three months pre-intervention, from April 2026 to June 2026, and three months post-intervention, from July 2026 to September 2026). The assistant will conduct active searches for potential participants. Eligible patients will be invited to provide informed consent.

**Initial interview.** The interview will start with a questionnaire adapted from Lichen et al. (2021) and McCusker et al. (2018) [2,3].

**Questions adapted from Lichen et al. (2021)**

| *Question* | *Response options* |
|---|---|
| *During your stay in the ED, what was your average comfort level?* | Very comfortable; Comfortable; Somewhat comfortable; Somewhat uncomfortable; Uncomfortable; Very uncomfortable |
| *As a person in the emergency department, do you feel you were offered care specific to your individual needs (I.e. needs related to communication, hearing, vision, etc.)?* | Strongly agree; Somewhat agree; Neither agree nor disagree; Somewhat disagree; Strongly disagree |
| Additional questions in the post-implementation phase | |
| *Did you use any of the items or services offered in the Geriatric Cart?* | Yes; No |
| *Knowing that these items or services were available, even if I did not use them, made me feel more comfortable* | Strongly agree; Somewhat agree; Neither agree nor disagree; Somewhat disagree; Strongly disagree |
| *The items I requested from the comfort cart made me feel more comfortable during my visit today* | Strongly agree; Somewhat agree; Neither agree nor disagree; Somewhat disagree; Strongly disagree |
| *Having the comfort cart available improved my overall experience today* | Strongly agree; Somewhat agree; Neither agree nor disagree; Somewhat disagree; Strongly disagree |
| *Having the comfort cart gave me more independence during my visit today* | Strongly agree; Somewhat agree; Neither agree nor disagree; Somewhat disagree; Strongly disagree |

**Questions adapted from McCusker et al. (2018)**

| *Question* | *Response options indicating no problem* | *Response options indicating problem* |
|---|---|---|
| *Do you feel that the ED staff did everything they could to relieve your pain or discomfort?* | Yes, definitely; No pain (not applicable) | No; Somewhat; Don't know |
| *Did the staff treat you with respect and dignity while you were in the ED?* | Yes, definitely | No; Somewhat; Don't know |

| Questions adapted from Lichen et al. (2021) | | |
|---|---|---|
| *Was your health problem explained to you in a way that you could understand?* | Yes, definitely; It is still being investigated; It was not necessary | No; Somewhat; Don't know |
| *Did someone explain to you the tests you had to have?* | Yes, every time; It was not necessary | No; Sometimes; Don't know |
| *If your family member wanted to talk to a doctor, did they have the opportunity to do so?* | Yes; I do not have a family member present; It was not necessary | No; Don't know |
| *Before you left, did a staff member provide your family member or friend, who was with you in the ED, all the information they needed to help you recover?* | Yes, definitely; It is still being investigated; I do not have a family member present; It was not necessary | No; Don't know |
| *Were you told what signs related to your health problem to watch out for when you get home?* | Yes, definitely; It is still being investigated; My family member was informed; It was not necessary | No; Somewhat; Don't know |
| *Were you given advise about resuming your normal daily activities?* | Yes, definitely; It is still being investigated; My family member was informed; It was not necessary | No; Somewhat; Don't know |

The following data will then be collected:

| Demographics | Full name, medical record number, ED visit number, date of birth, postal code, sex, date and time of ED admission |
|---|---|
| Source of information | If not the patient, the source's full name, date of birth, sex, and relationship to the patient will be recorded |
| Sensory deficits | Presence and type of visual and/or hearing impairments |
| History of falls | Number of falls in the past year, circumstances surrounding the falls |
| Comorbidity | Assessed using the Charlson Comorbidity Index (CCI) [12–15] |
| Risk of falls | Assessed using the Memorial Emergency Department Fall Risk Assessment Tool (MEDFRAT) and the Carpenter instrument [16,17] |
| Frailty | Assessed using the Clinical Frailty Scale (CFS) version 2.0 [18,19] |
| Clinical severity | Assessed using the National Early Warning Score (NEWS) 2 [20,21] |
| Mood | Assessed using the 15-item Geriatric Depression Scale (GDS-15) [22] |
| Geriatric vulnerability | Assessed using the PRO-AGE Score [10] |
| Acute mental status change (delirium) | Assessed using the Delirium Triage Screen (DTS) and Brief Confusion Assessment Method (bCAM) [23] |
| Quality of life | Assessed using the Euro Quality of Life Instrument – 5D (EQ-5D) [24] |
| Functionality | Assessed using the Katz Index, Barthel Index, and Lawton Scale [25–27] |
| Cognitive performance | Assessed using the 10-Point Cognitive Screener (10-CS) [28] |

**Medical and multidisciplinary staff interview.** A trained research assistant will interview medical and multidisciplinary staff concurrently with patient recruitment. The interview will follow the standardization of the original comfort cart validation study [3], using a Likert scale to answer the following questions.

| Questions adapted from Lichen et al. (2021) | |
|---|---|
| *Question* | *Response options* |
| *During the patient's stay in the ED, what was his/her average level of comfort?* | Very comfortable; Comfortable; Somewhat comfortable; Somewhat uncomfortable; Uncomfortable; Very uncomfortable |
| *I feel that age-friendly care specific to the geriatric patient's needs was provided* | Strongly agree; Somewhat agree; Neither agree nor disagree; Somewhat disagree; Strongly disagree |
| *I currently have access to non-pharmacological resources to provide comfort for geriatric patients* | Strongly agree; Somewhat agree; Neither agree nor disagree; Somewhat disagree; Strongly disagree |

| Questions adapted from Lichen et al. (2021) | |
|---|---|
| *Offering non-pharmacological resources to my patient increases my work burden by too much* | Strongly agree; Somewhat agree; Neither agree nor disagree; Somewhat disagree; Strongly disagree |
| Additional questions in the post-implementation phase | |
| *The Geriatric Cart and Menu of Items are a non-pharmacologic way to provide comfort* | Strongly agree; Somewhat agree; Neither agree nor disagree; Somewhat disagree; Strongly disagree |
| *The Geriatric Cart and Menu of Items are an important part of a comprehensive approach for improving patient satisfaction* | Strongly agree; Somewhat agree; Neither agree nor disagree; Somewhat disagree; Strongly disagree |
| *The Geriatric Cart and Menu of Items help increase my ability to care for older adults in a more compassionate way* | Strongly agree; Somewhat agree; Neither agree nor disagree; Somewhat disagree; Strongly disagree |
| *Having a hearing amplifier, reading glasses or magnifying glasses available for use helps patients feel more oriented/ less confused* | Strongly agree; Somewhat agree; Neither agree nor disagree; Somewhat disagree; Strongly disagree |

**Follow-up interview.** A second research assistant, blinded to the initial assessment and ED care details, will conduct a phone interview 30 days (± 2 days) post-initial assessment. This interview will assess the following:

| ED readmissions | Number of ED visits since the initial visit |
|---|---|
| **Falls** | Number of falls since the initial visit |
| **Delirium incidence** | Assessed using the Family Confusion Assessment Method (FAM-CAM) [29] administered to a family member or caregiver |
| **Mood** | Assessed using the GDS-15 |
| **Quality of life** | Assessed using the EQ-5D |
| **Functionality** | Assessed using the Katz Index, Barthel Index, and Lawton Scale |
| **Cognitive performance** | Assessed using the 10-CS |
| **Mortality** | If the patient has passed away, the date and cause of death will be recorded |

**Chart review and administrative data.** Length of hospital stay and hospital costs will be extracted from the electronic medical record and HSL's Business Intelligence (B.I.) administrative databases.

## Intervention

The comfort menu and cart will be implemented after the pre-intervention data collection phase is completed. The menu will list the items available on the cart and will be presented to eligible patients in printed or digital format (on a tablet). The cart will be placed in an easily accessible area within the ED. Approximated retail prices have been previously reported [3].

A one-month dedicated rollout period will precede the commencement of post-intervention participant enrollment. This phase is designed for comprehensive staff training on the proper use and components of the comfort menu and cart, including familiarization with all available items and menu options. Additionally, this period will ensure all necessary equipment is adequately prepared and any potential logistical challenges associated with the cart's implementation are identified and resolved, thereby optimizing the fidelity and effectiveness of the intervention before data collection for the post-intervention group begins.

Comfort Cart Contents:

• Hot and cold packs;

• Extra blanket and pillow;

• Face towel;

- Gloves and hat;

- Personal hygiene kit (toothbrush, toothpaste, cotton swabs, sleep mask, comb, deodorant, hand lotion);

- Items for distraction (magazines, word search puzzles, playing cards, coloring pages);

- Items to enhance communication (reading glasses, magnifying glass, penlight, cell phone charger, notepad, sound amplification device).

    Comfort Menu Options:

- Physiotherapy assessment;

- Conversation with chaplain, concierge, or social worker;

- List of resources for older adults needing community services;

- Home care assistance.

## Data management and statistical analysis

All study data will be collected and managed using Research Electronic Data Capture (REDCap) [30], a secure web-based application. Research assistants will use tablets with online access to the REDCap database for direct data entry and informed consent procedures.

For the primary objective evaluating the impact of the comfort menu and cart on the experience of older patients we will use the questionnaire adapted from Lichen et al. (2021) and McCusker et al. (2018). For the secondary objectives evaluating the impact of a comfort menu and cart on the experience of medical and multidisciplinary staff involved in caring for older patients treated in the ED, we will use the the questionnaire adapted from Lichen et al. (2021).

Data analysis will be performed using Stata version 17 (StataCorp, College Station, TX). All statistical tests will be two-tailed with an alpha error of 0.05. Numerical variables will be reported as means and standard deviations or medians and interquartile ranges (IQR) depending on their distribution, which will be assessed visually and using the D'Agostino-Pearson test for normality. Categorical variables will be described as absolute counts and proportions. Numerical variables will be compared using Student's t-test or ANOVA for normal distributions and Wilcoxon or Kruskal-Wallis tests for non-normal distributions. As appropriate, categorical variables will be compared using the chi-square test or Fisher's exact test. Analyses will be adjusted to account for potential confounding variables, such as baseline frailty, cognitive impairment, and clinical severity. Missing data will be handled though mixed effects model.

All relevant data from this study will be made available upon study completion.

## Patient and public involvement

While patients or the public were not directly involved in the drafting of this specific study protocol, the study's core components are founded on patient-centered principles. The comfort cart intervention is adapted from the work of Lichen et al. (2021), which was developed to address needs identified by older adults and included the patient's voice in the selection of the items for the comfort cart. Furthermore, the primary patient experience outcome measures are adapted from the questionnaire developed by McCusker et al. (2018), which was created through direct qualitative interviews with older adults about their emergency department experiences. We will disseminate the study findings to patient communities and participants upon completion of the research.

## Ethics and dissemination

Ethical approval for this study has been granted by the Institutional Review Board of HSL (*Comitê de Ética em Pesquisa do Hospital Sírio Libanês/ Sociedade Beneficente de Senhoras*): certificate nº 85870125.4.0000.5461, decision nº

7.436.261, dated March 12 2025. All participants, or their legal representatives for those with cognitive impairment, will provide written informed consent before any study procedures are initiated.

Given that the study employs a quasi-experimental design, with no alteration to the participants' clinical procedures, and includes participants both before and after the intervention, the informed consent form will not disclose details of the intervention in order to mitigate information bias. The purpose is to ensure that participants are unaware of the intervention studied, thereby preventing their perception of care from being affected by a perceived comfort, either absent or present, depending on their study phase.

The results will be shared with the academic community through peer-reviewed publications and presentations at relevant conferences to inform future clinical practice and research.

## Expected results

We expect to demonstrate the positive impact of implementing the comfort menu and cart in the ED on patient-centered outcomes. We anticipate improvements in the experience of older patients and medical and multidisciplinary staff, and we hope to find improvement in other exploratory outcomes, such as length of hospital stay, hospital costs, readmissions, falls, delirium incidence, quality of life, functionality, and cognitive performance.

| Year | 2025 | | | | | | | | | | | | 2026 | | | | | | | | | | | | 2027 | | | | | |
|---|---|---|---|---|---|---|---|---|---|---|---|---|---|---|---|---|---|---|---|---|---|---|---|---|---|---|---|---|---|---|
| **Month** | 1 | 2 | 3 | 4 | 5 | 6 | 7 | 8 | 9 | 10 | 11 | 12 | 1 | 2 | 3 | 4 | 5 | 6 | 7 | 8 | 9 | 10 | 11 | 12 | 1 | 2 | 3 | 4 | 5 | 6 |
| **Ethics committee and institutional submission and processing** | X | X | X | X | X | X | | | | | | | | | | | | | | | | | | | | | | | | |
| **REDCap project development** | | | | | | | | | | X | X | X | X | X | X | | | | | | | | | | | | | | | |
| **Hiring and training of research assistant** | | | | | | | | | | X | X | X | X | X | X | | | | | | | | | | | | | | | |
| **Pre-intervention case inclusion/recruitment** | | | | | | | | | | | | | | | | X | X | X | | | | | | | | | | | | |
| **Intervention structuring – comfort cart (rollout period)** | | | | | | | | | | | | | | | | | | X | | | | | | | | | | | | |
| **Implementation of the comfort cart** | | | | | | | | | | | | | | | | | | | X | X | X | | | | | | | | | |
| **Post-intervention case inclusion/recruitment** | | | | | | | | | | | | | | | | | | | X | X | X | | | | | | | | | |
| **Follow-up interview** | | | | | | | | | | | | | | | | | | | X | X | X | X | X | X | | | | | | |
| **Data processing and statistical analysis** | | | | | | | | | | | | | | | | | | | | | | X | X | X | X | | | | | |
| **Preparation of reports and articles for publication** | | | | | | | | | | | | | | | | | | | | | | | | | | | X | X | X | X |

## Discussion

Our study focuses on implementing and evaluating a comfort menu and cart, a non-pharmacological, innovative intervention designed to enhance the experience of older adults in the ED. This approach aligns with international guidelines emphasizing the importance of creating a supportive and age-friendly ED environment. By providing older patients access to items that promote comfort, communication, and engagement, we aim to improve their overall experience, potentially leading to better clinical outcomes.

While previous studies have explored similar interventions in other acute care settings, our study will provide valuable evidence specific to the ED context. Furthermore, our study will be the first to assess the impact of this intervention on a wide range of patient-centered outcomes measured up to 30 days after the initial ED visit. This longitudinal perspective will provide a more comprehensive understanding of the potential benefits of the intervention. Our study will also advance the growing body of literature on non-pharmacological interventions for older adults in acute care settings. By

demonstrating the feasibility and effectiveness of a relatively simple and low-cost intervention, our study can encourage wider adoption of similar approaches in EDs and other healthcare settings.

Our study has limitations that should be acknowledged. Firstly, it will be conducted in a single tertiary philanthropic hospital in Brazil, which may limit the generalizability of the findings to other ED settings, particularly those with different resource availability or patient populations. Secondly, while allowing for a comparison of outcomes before and after the intervention, the before-and-after design, is not as robust as a randomized controlled trial in establishing causality. Thirdly, the reliance on patient and staff self-reported data for assessing experience and comfort may introduce some information bias. However, we will use validated questionnaires and scales to mitigate this potential limitation. Finally, the 30-day follow-up period may not be sufficient to understand potential longer-term effects of the intervention. Future studies with longer follow-up periods are warranted.

## Conclusion

Despite these limitations, our study will provide cutting-edge evidence on the impact of a comfort menu and cart on the experience of older adults in the ED. By comprehensively assessing patient experience, staff experience, and clinical outcomes, our findings will contribute to developing and implementing effective strategies to improve the care of the growing number of older patients requiring acute care at the ED for unplanned conditions. We anticipate that our study will demonstrate the feasibility of this innovative, non-pharmacological intervention, encouraging wider adoption of similar approaches to enhance the well-being of older adults in fast-paced acute care settings.

## Supporting information

**S1 Checklist. SPIRIT 2025 editable checklist.**
(DOCX)

**S1 File. Study protocol approved by ethics committee.**
(PDF)

**S2 File. Study protocol approved by ethics committee_English.**
(PDF)

## Acknowledgments

The authors thank those who will support this study, particularly the Instituto de Ensino e Pesquisa, Hospital Sírio-Libanês, São Paulo.

## Author contributions

**Conceptualization:** Silvia A. Ferreira, Fernanda Bellolio, Susan M. Bower, Marlon R. Aliberti, Pedro Kallas Curiati.

**Data curation:** Marlon R. Aliberti, Pedro Kallas Curiati.

**Formal analysis:** Pedro Kallas Curiati.

**Funding acquisition:** Pedro Kallas Curiati.

**Investigation:** Pedro Kallas Curiati.

**Methodology:** Silvia A. Ferreira, Susan M. Bower, Marlon R. Aliberti, Thiago J. Avelino-Silva, Christian V. Morinaga, Pedro Kallas Curiati.

**Project administration:** Christian V. Morinaga, Pedro Kallas Curiati.

**Resources:** Luis Fernando Penna, Magali A. L. Marion, Christian V. Morinaga, Pedro Kallas Curiati.

**Supervision:** Fernanda Bellolio, Luis Fernando Penna, Magali A. L. Marion, Marlon R. Aliberti, Christian V. Morinaga, Pedro Kallas Curiati.

**Writing – original draft:** Silvia A. Ferreira, Pedro Kallas Curiati.

**Writing – review & editing:** Fernanda Bellolio, Susan M. Bower, Luis Fernando Penna, Magali A. L. Marion, Marlon R. Aliberti, Thiago J. Avelino-Silva, Christian V. Morinaga.

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
