## [Decision Letter · Decision Letter 0]

4 Nov 2025

Dear Dr. Curiati,

We look forward to receiving your revised manuscript.

Kind regards,

Emma Campbell, Ph.D

Staff Editor

PLOS ONE

2. In the online submission form, you indicated that [No datasets were generated or analysed during the current study. Following study completion, de-identified participant data and the full study protocol will be made available from the corresponding author upon reasonable request.].

4. Please ensure that you refer to Figure 1 in your text as, if accepted, production will need this reference to link the reader to the figure.

Additional Editor Comments (if provided):

Reviewers' comments:

Reviewer's Responses to Questions

**Comments to the Author**

1. Does the manuscript provide a valid rationale for the proposed study, with clearly identified and justified research questions?

Reviewer #1: Yes

2. Is the protocol technically sound and planned in a manner that will lead to a meaningful outcome and allow testing the stated hypotheses?

Reviewer #1: Yes

3. Is the methodology feasible and described in sufficient detail to allow the work to be replicable?

Reviewer #1: Yes

4. Have the authors described where all data underlying the findings will be made available when the study is complete?

Reviewer #1: Yes

5. Is the manuscript presented in an intelligible fashion and written in standard English?

Reviewer #1: Yes

You may also provide optional suggestions and comments to authors that they might find helpful in planning their study.

Reviewer #1: This paper is a description of the intended protocol for a pre/post intervention study to deploy a care cart for older patients in the ED at a single hospital in Brazil. It explains how patients will be enrolled, what questions will be assessed, how participants will be followed up, and what items will be contained in the care cart. The power analysis is appropriate for the study, feasibility information is provided about potential enrollment, and the proposed statistical analysis plans are appropriate.

The only critique/comment I have about the study design pertains to the timing of the roll-out of the cart. The study plans to enroll the pre group over a period of 3 months (which seems appropriate based on feasibility numbers and power calculations). The study then plans to immediately enroll the batch of post participants. This leaves no time for roll-out/equipping of the cart and training in proper use of the cart. It also doesn't allow for any hiccups in the cart roll-out process. This reviewer would suggest including a roll-out phase for the care cart of a month before enrolling participants in a study which will use the cart. This will allow for any potential problems to be resolved before participants are enrolled.

**Do you want your identity to be public for this peer review?** For information about this choice, including consent withdrawal, please see our Privacy Policy

Reviewer #1: No

---

## [Author Response · Author response to Decision Letter 1]

4 Nov 2025

Response to Reviewers' Comments

We thank the reviewer for the insightful feedback and constructive suggestions, which have greatly helped us improve the quality of our manuscript. Below, we provide a point-by-point response to each comment, detailing how the revisions have been incorporated into the updated submission.

Reviewer #1

Comment: "The only critique/comment I have about the study design pertains to the timing of the roll-out of the cart. The study plans to enroll the pre group over a period of 3 months (which seems appropriate based on feasibility numbers and power calculations). The study then plans to immediately enroll the batch of post participants. This leaves no time for roll-out/equipping of the cart and training in proper use of the cart. It also doesn't allow for any hiccups in the cart roll-out process. This reviewer would suggest including a roll-out phase for the care cart of a month before enrolling participants in a study which will use the cart. This will allow for any potential problems to be resolved before participants are enrolled."

Response: We thank the reviewer for this crucial suggestion. We fully agree that a dedicated rollout period is essential to ensure the seamless implementation of the comfort menu and cart, allowing for proper staff training and resolution of potential logistical challenges prior to the post-intervention data collection. We have incorporated this recommendation into the "Intervention" section of the manuscript (page 17, lines 13-21) to reflect this important preparatory phase:

"A one-month dedicated rollout period will precede the commencement of post-intervention participant enrollment. This phase is designed for comprehensive staff training on the proper use and components of the comfort menu and cart, including familiarization with all available items and menu options. Additionally, this period will ensure all necessary equipment is adequately prepared and any potential logistical challenges associated with the cart's implementation are identified and resolved, thereby optimizing the fidelity and effectiveness of the intervention before data collection for the post-intervention group begins."

---

## [Editor Report · Decision Letter 1]

19 Nov 2025

Enhancing the Emergency Department Experience for Older Adults: Study Protocol for the Implementation of a Comfort Menu and Cart

PONE-D-25-42092R1

Dear Dr. Curiati,

We’re pleased to inform you that your manuscript has been judged scientifically suitable for publication and will be formally accepted for publication once it meets all outstanding technical requirements.

As a Peer-Reviewed Funded Protocol, your submission is eligible for expedited review by in-house editors. Based on our evaluation, we are satisfied that your manuscript meets our publication criteria for Study Protocols, and is therefore considered to be suitable for publication subject to final journal requirements.

Kind regards,

Emma Campbell, Ph.D

Staff Editor

PLOS ONE
---

## [Editor Report · Acceptance letter]

PONE-D-25-42092R1

PLOS ONE

Dear Dr. Curiati,

I'm pleased to inform you that your manuscript has been deemed suitable for publication in PLOS ONE. Congratulations! Your manuscript is now being handed over to our production team.

Kind regards,

on behalf of

Dr Emma Campbell

Staff Editor

PLOS ONE